# Late Pleistocene temperature patterns in the Western Palearctic: insights from rodent associations compared with General Circulation Models

5 Aurélien Royer<sup>1,\*</sup>, Julien Crétat<sup>1</sup>, Rémi Laffont<sup>1</sup>, Sara Gamboa<sup>2</sup>, Belén Luna<sup>3</sup>, Iris Menéndez<sup>4</sup>, Benjamin Pohl<sup>1</sup>, Sophie Montuire<sup>1,5</sup>, Manuel Hernandez Fernandez<sup>6,7</sup>

- <sup>2</sup> Centro de Investigación Mariña, Universidade de Vigo, MAPAS Lab, Campus Lagoas-Marcosende, Fonte das Abelleiras s/n, 36310 Vigo, Spain.
  - <sup>3</sup> Departamento de Ciencias Ambientales, Facultad de Ciencias Ambientales y Bioquímica, Universidad de Castilla-La Mancha, Carlos III s/n, 45071 Toledo, Spain.
  - <sup>4</sup> Museum für Naturkunde, Leibniz Institute for Research on Evolution and Biodiversity at the Humboldt University Berlin, 10115 Berlin, Germany.
- <sup>5</sup> Ecole Pratique des Hautes Etudes, PSL University, Paris, France
  - <sup>6</sup> Departamento de Geodinámica, Estratigrafía y Paleontología, Facultad de Ciencias Geológicas. Universidad Complutense de Madrid. José Antonio Nováis 12, 28040 Madrid, Spain.
  - <sup>7</sup> Departamento de Cambio Medioambiental, Instituto de Geociencias (UCM, CSIC). Severo Ochoa 7, 28040 Madrid, Spain.
  - \* corresponding author: aurelien.royer@u-bourgogne.fr

Abstract word count: 384

Main text word count: 10713

Figures: 13
Tables: 2

Appendices: 6

Supplementary material: 1

<sup>&</sup>lt;sup>1</sup> Biogéosciences, UMR 6282 CNRS, EPHE, Université Bourgogne Europe, 6 Boulevard Gabriel, 21000 Dijon, France.

**Abstract.** Since rodent fossils are preserved in many low and high latitude archaeological and paleontological sites from a wide variety of environments, their associations are a commonly useful proxy for inferring past local climate and environmental conditions. Such a frequent and widespread geographic distribution can help us to better understand past climate evolution by providing access to high spatiotemporal resolution at large geographical scales. The aim of this paper is to develop an approach to generate continental scale temperature maps based on rodent associations and to assess their reliability compared to state-of-the-art General Circulation Models (GCMs). We used the Bioclimatic Analysis, based on fossil and modern rodent associations, to infer climate zone distribution and local temperatures (Mean Annual Temperature, Mean Temperature of the Warmest month and the Mean Temperature of the Coldest month), at the western Palearctic (Europe, Middle East and North Africa) for six different periods; LGM, Heinrich Stadial, Bølling, Allerød, Younger Dryas and presentday conditions. The Bioclimatic Analysis is combined with a spatial generalized linear mixed model to interpolate these surface temperatures across the western Palearctic. We show that the spatial patterns in Mean Annual Temperature and Mean Temperature of the Warmest and Coldest months are very similar between our interpolations and GCMs for both present-day and LGM conditions, but the rodent-based approach provides slightly cooler LGM estimations in western Europe and warmer in eastern Europe. Throughout the Late Glacial oscillations, the rodent-based model infers globally small variations in Mean Annual Temperature and Mean Temperature of the Warmest months and slightly larger changes in Mean Temperature of the Coldest months. Nonetheless, some events show weak but significant regional variations depending of the events and the climate variable. For instance, the most important shifts in mean annual temperature between Allerød and Younger Dryas are observed in northwestern regions. Northeastern regions, on the other hand, experienced relatively stable mean annual temperature, although they did experience considerable warming of the warmest month and cooling of the coldest month. Minor discrepancies appear between GCMs and the rodent-based model, the latter showing colder temperature in northwestern Europe, hence a differential west-east gradient in ice-sheet influence. Our results demonstrate that rodent associations are robust proxies for reconstructing and regionalizing past temperatures at broad scales, offering a readily reproducible approach to be reimplemented in future studies incorporating new rodent data.

**Keywords**: rodent-based reconstructions, temperature patterns, Last Glacial Maximum, Late Glacial, Heinrich Stadial, Bølling-Allerød Interstadial, Younger Dryas,

#### 1 Introduction

Understanding past climate changes is necessary for interpreting the evolution of living organisms and their communities in their historical environmental context. Past climate information comes from climate proxy record (in the form of geochemical, physical, geological or biological data) obtained from a natural archive (Markwick, 2007) and/or from General Circulation Models based on thermo-dynamic laws (Randall, 2000). Each proxy has its own set of advantages and limitations. Some are directly correlated to climatic variables (e.g. some isotopes), while others reflect their indirect influence (e.g. pollen, invertebrate or vertebrate associations). Nonetheless, all are complementary, because each proxy provides a piece of information about the global system for a specific sphere (e.g. hydrosphere, biosphere), on particular geographical and time scales, and at a certain resolution. Paleoclimatic inference methods on different climate proxies are constantly improved to overcome their limitations and to provide refined past climate change information. As any single proxy still provides a unique history of the past environment and climate, many studies have emphasized the need for multi-proxy approaches and/or comparisons between data from different proxy natures (e.g. Allen et al., 2008) to have a better understanding of paleoclimate proxy records and climate variations.

Among the many natural archives (e.g. speleothems, chironomids, pollen) available to approximate past climate changes, fossil remains of small mammals (e.g. Rodentia and /or Eulipotyphla) have been used for a long time, because they present several advantages:

- 1) their large taxonomic diversity is associated with a wide range of ecological and biological traits that ensure their ubiquitous occurrence in all modern continental biotopes (e.g. Quéré & Le Louarn, 2011; Wilson et al., 2017; Wilson & Mittermeier, 2018);
  - 2) their small size, associated with high reproductive rates and fast evolutionary change, makes them particularly sensitive to changes in climate and habitat, which constrain their geographical distributions, biotic interactions and, ultimately, community organization (Wolff & Sherman, 2008);
- 3) they are regularly found in archaeological and paleontological sites (e.g. Hinton, 1926; Chaline, 1972; Daams et al., 1988; Martin et al., 2000; van den Hoek Ostende, 2003; Cuenca-Bescós et al., 2010; Furió et al., 2011; Minwer-Barakat et al., 2012; Erbajeva & Alexeeva, 2013; Royer et al., 2016; Harzhauser et al., 2017; Markova et al., 2019; Flynn et al., 2023).

As a consequence, small vertebrates and in particular rodents have been recognized as good proxy and regularly used for inferring past local and continental environmental and climatic conditions (e.g. Simar-Pellissier, 1966; Chaline, 1972; Tchernov, 1975; Avery, 1987; van der Meulen & Daams, 1992; Montuire & Desclaux, 1997; Montuire & Marcolini, 2002; Legendre et al., 2005; Hernández Fernández et al., 2007; García-Alix et al., 2008; Cuenca-Bescós et al., 2009; López-García et al., 2010; Royer et al., 2013, 2014; Rofes et al., 2015; Royer et al., 2016; Fernández-García et al., 2016; Mansino et al., 2016; Jeannet, 2018; Stoetzel et al., 2019; López-García et al., 2021; Royer et al., 2021; Linchamps et al., 2023; García-Morato et al., 2024; Piñero et al., 2024; Schürch et al., 2024).

In parallel, many different fully coupled General Circulation Models (GCMs) have been developed and used over the past decades to understand past, present-day and future climate variabilities and Earth system changes (e.g. Sloan et al., 1996; Prentice et al., 1998; Salzmann et al., 2008; Haywood et al., 2009; Henrot et al., 2017; Beyer et al., 2020; Kageyama et al., 2020; Kageyama & Paillard, 2021), and to explore their influence on human communities and environment changes (e.g. Allen et al., 2010; Burke et al., 2021; Leonardi et al., 2023; Albouy et al., 2024). GCMs use the equations of fluid thermodynamics and integrate biotic and abiotic variables to resolve the space-time changes and dynamics of the Earth climate system. GCMs provide numerical climate information at the global scale, and are therefore not limited by the uneven distribution of sample locations and taphonomic biases. Comparing results from different 100 climate proxies, such as small mammals, to GCM outputs offers considerable potential for gaining a better understanding of the mechanisms driving climate variability (e.g. Emile-Geay et al., 2016; Carré et al., 2021). It also provides unique opportunities for evaluating model performance and proxy applicability (e.g. Kageyama et al., 2001; Jost et al., 2005; Allen et al., 2010; Latombe et al., 2018; Comas-Bru et al., 2019). Although small mammal associations are generally compared with other proxy data at a local scale, 105 such as pollen or stable isotope compositions, combining paleoclimate inferences based on their assemblages with GCM paleoclimate simulations has been rarely tested for refining environmental and chronological context (e.g. Terray et al., 2023). The integration of such data sources into a general approach to paleoclimatic research yields the potential to attain major advances in the understanding of 110 climate change at large spatial scales.

Rodent fossil remains are abundant at many sites throughout Europe, from the Mediterranean region to high latitudes. This abundancy allows us not only to use their assemblages to reconstruct local climate conditions, but also to access continental-scale paleoclimate reconstructions under a wide range of environments and time intervals (e.g. Hernández Fernández, 2006; Puzachenko & Markova, 2019, 2023). In this paper, we use rodent associations to infer surface temperature under different climate conditions. We compare their inferences with GCM outputs to better elucidate faunal community signatures and refine climate reconstructions. We focus on large spatial scales, over the western Palearctic (Europe, Middle East and North Africa), where a large amount of fossil data is available from the Last Glacial Maximum (LGM) to the end of the Late Glacial, from 23,000 to 11,700 years before present. This period of major climate changes is characterized by a succession of short warm (Greenland Interstadial) and cold (Greenland Stadial) events leading to the so-called Bølling, Allerød, and Younger Dryas pollen chronozones (e.g. Björck et al., 1998; Peyron et al., 2005; Rasmussen et al., 2014). These climatic events have strongly shaped Palearctic environments and faunal communities.

The aim of this study is twofold:

- First, we propose a new approach to interpolate surface temperature derived from species composition of rodent associations by combining the Bioclimatic Analysis (Hernández Fernández & Peláez-Campomanes, 2003, 2005; Royer et al., 2020) to a spatial generalized linear mixed model. This approach is used to reconstruct mean annual temperature (MAT), as well as mean temperature of the warmest (MTWA) and coldest months (MTCO), in the present-day context and for successive events within the LGM Late Glacial interval.
  - Second, we compare these reconstructions with GCM simulations integrated over the same time periods, to determine the degree of spatiotemporal agreement between those approaches.

#### 2 Data and method

140

# 2.1 Rodent associations from modern localities

A total of 157 modern localities covering northern Africa and western Eurasia (north of 25° N and west of 60° E) were used to reconstruct spatial variation in present-day climate (Figure 1; Table C1). Most of the sites (n=152) are located at low altitude (below 1000 m above current sea level) to focus on large-scale signal with no interference of local topography; only five localities are at an altitude higher than 1000 m above sea level. For each locality, we compiled a rodent faunal list (Table C2) by identifying all rodent species whose geographic range polygons, sourced from the IUCN Red List spatial data (IUCN, 2021), intersect with a 50 km radius buffer around the locality coordinates.

Figure 1: Geographical distribution of modern (circles) and fossil (triangles) rodent associations studied in this work. Pink circles show the modern assemblages used to calculate area-averaged MAT, MTWA and MTCO values.

# 2.2 Temporal framework and rodent associations from fossil localities

We focused on rodent associations from the Last Glacial Maximum (LGM) to the end of Late Glacial, *i.e.* between 23,000 and 11,700 years before present (BP), compiling a database that includes 279 associations coming from stratigraphic units in 168 archaeological and paleontological sites located from Morocco to the Ural mountains (Fig. 1, Table C2). These fossil associations do not represent a synchronous ensemble. To maximize fossil data availability, we grouped them into six time intervals that correspond to major climatic events and pollen chronozones: LGM (GS2.1bc: 23,000 – 18,000 years BP;

fossil associations), Heinrich Stadial 1 (GS2.1a: 18,000 – 14,900 BP; 93 ass.), Bølling (GI-1de: 14,900 – 13,900 BP; 47 ass.), Allerød (GI-1abc: 13,900 – 12,900 BP; 41 ass.) and Younger Dryas (GS-1: 12,900 – 11,700 BP; 33 ass.).

Reconstructing paleoenvironmental and paleoclimatic conditions from faunal remains requires specific analytical and taphonomical considerations (Lyman, 2019) to ensure high data quality. As such data is derived from biological communities, the quality of faunal data depends on accurate taxonomic identifications. Many species are indeed highly difficult to identify due to strong similarity, high morphological variability or extremely low local abundance (e.g. Nadachowski, 1982; Markova et al., 2010; Escudé et al., 2013; Navarro et al., 2018; Arbez et al., 2021; Stoetzel et al., 2023). Faunal remains primarily come from field data, which are regularly re-evaluated to verify the data integrity that may have gone unnoticed during the fieldwork or initial study, in relation to stratigraphic formation, palimpsest processes, and excavation unit definition (e.g. Lyman, 2008; Royer, 2014; Discamps et al., 2023), as well as bioturbation (e.g. Pelletier et al., 2017). To select the stratigraphic units, we ensured that the associated rodent assemblages were detailed and representative, maintaining good taphonomical integrity of small mammal remains within the archaeological and paleontological stratigraphic units. Sites that did not meet these criteria were not included, despite their contributions, such as the presence of the collared lemming (Dicrostonyx torquatus) in Croatia (Lenardić, 2013; Lenardić et al., 2018) or the striped field mouse (Apodemus agrarius) in France (Aguilar et al., 2008) during the LGM. Both are found in sites for which unfortunately the associated rodents are unknown. Since each fossil association should be included in a particular time interval, we selected mainly stratigraphic units associated with radiocarbon dates restricted to a single time interval (Table C2). Radiocarbon dates were calibrated at  $2\sigma$  (95.4%) using the *oxcAAR* v1.1.1 R package (Hinz et al., 2018) and IntCal20 calibration curve (Reimer et al., 2020).

### 2.3 The Bioclimatic Analysis

Many methods have been developed to estimate climatic variables from rodent communities (e.g. Hokr, 180 1951; van de Weerd & Daams, 1978; Daams & van der Meulen, 1984; Chaline & Brochet, 1989; Sesé, 1991; Montuire et al., 1997; van Dam, 2006; Jeannet, 2010; Fagoaga et al., 2019). We used here the Bioclimatic Analysis approach, originally developed by Hernández Fernández (2001) and Hernández Fernández & Peláez-Campomanes (2003, 2005) and updated by Royer et al. (2020). This approach, detailed in these studies, is a commonly used technique in paleoclimatic studies of fossil sites (e.g. 185 Hernández Fernández, 2006; Hernández Fernández et al., 2007; Pérez-Crespo et al., 2013; Socha, 2014; Laplana et al., 2016; Piñero et al., 2016; López-García et al., 2017; Berto et al., 2019b; Fernández-García et al., 2020; Lemanik et al., 2020; Álvarez-Vena et al., 2021; Izvarin et al., 2022; Jovanović et al., 2022; Luzi et al., 2022b; Arbez et al., 2023; Domínguez-García et al., 2023, 2024; Lebreton & López-García, 2023; Rey-Rodríguez et al., 2024; Stoetzel et al., 2025). It skilfully assesses multiple past climate 190 variables (e.g. temperature and precipitation) from mammal communities and provides reliable estimations as well as their associated uncertainties.

The original models for the Bioclimatic Analysis (Hernández Fernández & Peláez-Campomanes, 2003, 2005) were based on 50 modern localities at global scale and successfully validated by using an additional set of different localities from the ones used to develop them. The current models constructed by Royer

et al. (2020) were based on a new set of 49 modern communities distributed throughout the Palearctic, in order to be representative of the different climate zones (seven localities for seven climate zones). In that paper, the models were also validated based on Leave-One-Out Cross Validation (LOOCV).

This approach is based on the bioclimatic breath of each species (assessed through the Climatic Restriction Index, CRI; Hernández Fernández, 2001), which is based on the presence of each species in different climate zones (Table 1), as defined by the combination between monthly precipitation and temperature mean values (Walter, 1970). For this study, we added CRI values for more than one hundred species not defined in the original model, including some extinct species and some "chimeric" species, which are taxa only identified at genus levels or not distinguished from morphological sibling species. These CRI values were defined as in the original model, following the recently updated taxonomic and geographical range information (Wilson et al., 2017). All these added CRI values are detailed in the Table C3.

| Climate                    |           | Zonobiome (mainly vegetation type) |  |  |  |
|----------------------------|-----------|------------------------------------|--|--|--|
| I Equatorial               |           | Evergreen tropical rain forest     |  |  |  |
| II Tropical with sumr      | mer rains | Tropical deciduous woodland        |  |  |  |
| II/III Transition tropical | semiarid  | Savanna                            |  |  |  |

| Ш    | Subtropical arid                                 | Subtropical desert                 |
|------|--------------------------------------------------|------------------------------------|
| IV   | Subtropical with winter rains and summer drought | Sclerophyllous woodland-shrubland  |
| V    | Warm-temperate                                   | Temperate evergreen forest         |
| VI   | Typical temperate                                | Nemoral broadleaf-deciduous forest |
| VII  | Arid-temperate                                   | Steppe to cold desert              |
| VIII | Cold-temperate (boreal)                          | Boreal coniferous forest (taiga)   |
| IX   | Polar                                            | Tundra                             |

Table 1: Climate zone typology (from Walter, 1970) and its relationships with world vegetation types.

- The bioclimatic spectrum of a locality consists of ten bioclimatic components (BC) values (Hernández Fernández, 2001), calculated on the basis of the CRI values for each climate zone across the entire faunal assemblage and detailed in the Table C2. The use of BCs helps to address the problem of classifying paleoecological records by reducing the number of entities considered (only 10 BCs) and by providing an ecological basis for treating mammals from different regions in a compatible way.
- The Bioclimatic Analysis is composed of two parts. The first one relies on linear discriminant functions deduced from the bioclimatic spectra of modern mammalian communities. These linear discriminant functions are subsequently used to classify additional observations (extinct communities in our case) in each climatic zone, with an associated posterior probability (Hernández Fernández & Peláez-Campomanes, 2003). We used posterior probability values to assess the robustness of the climate classifications obtained by the discriminant functions, and considered robust probabilities above 0.95. A prediction error was estimated around 12% for estimating bioclimatic zone with linear discriminant functions on rodent communities (Royer et al., 2020). The second part of the Bioclimatic Analysis is built from transfer functions by means of multiple linear regression analyses of climatic parameters and modern

bioclimatic spectra. Most predictive equations generated by multiple linear regressions for each climatic factor, produced highly significant determination coefficients (Hernández Fernández & Peláez-Campomanes, 2005), and rarefaction analysis revealed these new models to be reliable even when a substantial percentage of species from the original community was removed (Royer et al., 2020). These models are ultimately used to infer climatic variables for additional observations (*i.e.* extinct communities). Although the Bioclimatic Analysis gives the possibility to estimate eleven climatic variables from fossil faunal assemblages (Hernández Fernández & Peláez-Campomanes, 2005; Royer et al., 2020), in this paper, we focus only on three of them: the Mean Annual surface Temperature (MAT), the Mean surface Temperature of the WArmest month (MTWA) and the Mean surface Temperature of the COldest month (MTCO), which are characterized by coefficients of determination of 0.94, 0.92 and 0.85, respectively (Royer et al., 2020).

## 2.4 Building interpolation maps

Once temperature data for individual localities were calculated, interpolation maps for the whole western Palearctic at each time interval were built using the 'isofit' and 'isoscape' functions from the R package *IsoriX* v.0.9.2 (Courtiol & Rousset, 2017; Courtiol et al., 2019), which is based on the *spaMM* v.4.4.16 (Rousset et al., 2016), *rasterVis* v.0.51.6 (Perpinan Lamigueiro et al., 2023) and *terra* v.1.7 (Hijmans et al., 2022) R packages. Interpolations of the three climatic variables estimated from rodent associations were inferred through a spatial generalized linear mixed model (GLMM), using latitude and longitude as covariates. Although the GLMM can include the effects of altitude, the number of high altitude fossil

sites is too limited per period to obtain sufficiently robust correlations to reproduce adiabatic effect, leading to erroneous models for geographical areas at middle and high altitudes. We therefore prefer to apply an average vertical drift rate, to reflect this effect in geographical and temporal terms. Elevation effects were accounted for by applying a temperature lapse rate of -0.5 °C/100 m on rodent association Bioclimatic Model results, and by reinjecting it after the spatial interpolation. The original functions of *IsoriX* were slightly modified to (1) better assess local prediction and uncertainties from the Bioclimatic Model and (2) account for ocean and ice sheet masks:

- 1) Initially, the *IsoriX* package required aggregated data at each locality to directly calculate the mean and variance before building the interpolation maps. We modified the 'isoscape' function to use the inferred data and predicted variance derived from applying the Bioclimatic Model instead of aggregated data. The original Bioclimatic Analysis script was also slightly modified to extract the standard error, the residual scale and the degree of freedom of the residuals generated for each locality, to calculate the variance of inferred data used by the *isoscape* function;
- 2) In order to obtain approximate boundaries for accessible areas, we generated ocean masks using mean bathymetry variations (Lambeck et al., 2014; Benjamin et al., 2017) of -130 m for the LGM, -120 m for the Heinrich Stadial 1, -100 m for the Bølling, -80 m for the Allerød and -60 m for the Younger Dryas. We also used ice sheet masks derived from the global ICE-6G\_C (VM5a) model (Argus et al., 2014; Peltier et al., 2015), with values at 21.0 ky for the LGM, 17.0 ky for the Heinrich Stadial 1, 14.5 ky for the Bølling, 13.5 ky for the Allerød and 12.5 ky for the Younger Dryas. The generated maps have a resolution of 0.1° x 0.1°.

The quality of the models produced and sites used was evaluated using standard deviation calculated from leave-one-out cross validation (LOOCV) to identify potential outliers to remove (Fig. B1). Three outliers (Bouârfa, Achgabat and Birjand) were detected and excluded for present day simulations, and one outlier impacting sizably temperature reconstructions was observed with the Younger Dryas Belarussian site of Volosovo.

## 2.5 General Circulation Model simulation of modern and past periods

An essential assessment for any method used in paleoclimatic reconstruction involves comparing inferred climate estimates with observed data. Even though the Bioclimatic Analysis has previously been tested on several occasions (Hernández Fernández & Peláez-Campomanes, 2003, 2005; Royer et al., 2020), we took the opportunity to evaluate its application by including the spatial dimension. To discuss similarities and differences between our rodent-based reconstructions and state-of-the-art GCMs from modern and past climatic conditions, we used ERA5 reanalysis, GCM simulations from the PMIP4 project and GCM simulations published by Beyer et al. (2020).

By combining model and large amounts of observations (e.g. weather stations, satellite and airborne measurements, buoys, radiosonde data), the state-of-the-art ERA5 global reanalysis (Hersbach et al., 2020) provides a spatially and temporally accurate climate dataset at  $0.25^{\circ}$  resolution, in particular in Europe (e.g. Johannsen et al., 2019). Here, we used ERA5 here as the reference for the modern climatic context for assessing the quality of our rodent-based model and GCMs. We have extracted monthly air temperature values 2-meters above the surface over land from the 1979-2008 period, from an area

bounded by the 25° N latitude and the 55° E longitude (which is therefore slightly less extensive than the rodent model).

The PMIP4 project regroups a large ensemble of GCM simulations built for different time periods, including the current present-day and the LGM (Kageyama et al., 2021). For the modern climatic context, we used six PMIP4 simulations for the 1976-2005 period: CCSM4 (Gent et al., 2011), CNRM-CM5 (Voldoire et al., 2013), IPSL-CM5A-LR (Dufresne et al., 2013), MIROC-ESM (Sueyoshi et al., 2013), MPI-ESM-P (Giorgetta et al., 2013) and MRI-CGCM3 (Yukimoto et al., 2012). For the LGM, we used the same six PMIP4 simulations plus the COSMOS-ASO (Stepanek & Lohmann, 2012) and FGOALS-g2 (Zheng & Yu, 2013) simulations. For conciseness, we consider the PMIP4 ensemble mean for spatial mapping.

Beyer et al. (2020) published 2m air temperature estimations combining medium and high resolution GCM (HadCM3 and HadAM3H) simulations and modern-era instrumental data (Beyer et al., 2020). This product, hereafter called Beyer2020, provides monthly bias-corrected 2 m air temperature and many other parameters at the 0.5° spatial resolution and 1,000-2,000-yr temporal resolution for the last 120,000 years, giving the possibility to compare the different periods of the Late Glacial. We extracted here the Beyer2020 simulations at -21.0, -17.0, -15.0, -13.0, -12.0 and 0 ky BP.

Finally, to quantify global changes between current and past climatic contexts, we selected 55 of the 157 localities as anchors to calculate average MAT, MTWA and MTCO values from our rodent-based reconstructions and the GCMs. These 55 localities cover the 35° N - 50° N / 15° W - 40° E domain (Fig. 1; Table C1), which was not impacted by ice sheet development (allowing a comparison between past

and present periods) and includes a large part of the archaeological and paleontological sites used in this 310 study, thus limiting the inferences extrapolation in data-free areas.

All the above analyses were performed using the R software (R core team, 2023). R scripts and raw data are available in Supplementary Material 1, in order to be fully reproducible.

#### 3 Results and discussion

#### 3.1 A simple approach from a complex system

Caution must always be exercised when working with reconstructions based on fossil faunal associations (e.g. Lyman, 2017). As any methodological approach based on biotic associations recorded in fossil sites, rodent-based paleoenvironmental reconstructions have limitations induced by multiple ecological assumptions. Knowing these assumptions remain essential to remember the limits of our interpretation, and using the Bioclimatic Analysis helps to partially overcome these limitations (Hernández Fernández, 2001; Hernández Fernández & Peláez-Campomanes, 2003, 2005; Royer et al., 2020).

The first limitation is the application of the principle of uniformitarianism, *i.e.* suggesting similar ecological tolerances between past and present (Lyman, 2017; Pineda-Munoz et al., 2021). This principle is the first to be used when studying faunal remains. We do know, however, that many species have a high degree of behavioral and ecological plasticity, or high level of endemism. Thus, some species may have changed and their ecological tolerances shifted. This is the reason why current paleoclimatic inference methods avoid considering only one species as a unique indicator of one specific climate, and

approaches use multiple taxa as a better means of reconstructing past environments. Using multiple species, as the Bioclimatic Analysis does, avoids then the overinterpretation related to one specific single taxon and allows then for a more nuanced understanding of climatic and environmental changes.

Second, the distribution pattern of modern biotic communities is the result of the combination of many factors (e.g. Jackson & Overpeck, 2000; Williams & Jackson, 2007; FAUNMAP, 1996), including geography (e.g. orography, refugia), biotic interactions (e.g. facilitation, competition) and history (e.g. climatic and/or geographic changes that cause populations expansion or contraction, isolation, extirpation/extinction and eventual speciation). This might lead to the development of past communities in which lived several species from contrasting environments, the so-called disharmonious faunas (Lundelius, 1989; Graham, 2005). The contrary effect is also found in modern assemblages, with some rodent species unexpectedly found in climatically very different environments, as the snow vole (Chionomys nivalis), which inhabits very high mountainous regions (e.g. the Alps and the Pyrenees) and lives equally in Mediterranean rocky inlets (Quéré & Le Louarn, 2011). In the same way, past refugia have played a fundamental role in the evolution of species, providing sites to survive extinction and to initiate colonization of new areas when environmental conditions became favorable (e.g. Tougard et al., 2008; Stewart et al., 2010; Royer et al., 2016; Baca et al., 2023a). The current period is a particular period when compared to all past periods, being the result of past history (Jackson & Overpeck, 2000; Graham, 2005; Williams & Jackson, 2007) plus anthropogenic activities. Anthropogenic activities strongly impact some rodents by offering opportunities to synanthropic species to develop, strongly modifying the realized niches of many other species (e.g. Pineda-Munoz et al., 2021), both by constraining or facilitating their expansion. For instance, anthropogenic activities in Europe have favoured species inhabiting open

environments through deforestation, such as the common vole, *Microtus arvalis* (e.g. Baca et al., 2023b), but have contracted significantly the range of the common hamster (*Cricetus cricetus*) in central Europe due to agricultural activities and landscape fragmentation (Surov et al., 2016). The Bioclimatic Analysis was established based on faunal associations from several typical distinct regions representing the different biomes found on Earth today and the bioclimatic characterization of the rodent faunas remains largely unbiased, since considering the ecology of each species at biome scale limits the species dependence effect to a particular environment. The combination of species interactions and their biomes also opens up the possibility of identifying special cases, related to taphonomical issues, possible particular ecotones that have occurred in the past, or biomes that were not similar to present ones.

Using faunal associations implies large prediction errors in climatic reconstructions since, like plants, rodents are directly and indirectly (through the resources they use) dependent on multiple "bioclimatic" variables (e.g. temperature fluctuation throughout the diurnal cycle, insolation, seasonal distribution of temperature and precipitation, snow cover, etc.) that influence their behavior and life cycle. For instance, lemmings inhabit environments where MTCO values can easily go down -15 °C, but this variable has a reduced impact on these arctic rodents, because they live under the snow. The critical factors for them are then the quality of the snow cover and the protection it provides from air surface temperature. More importantly, in comparison with plants, rodents belong to an upper trophic level, being also highly dependent on vegetation, which serves as food source and protection from climate and predators. Some species are very restrictive on one type of vegetation, such as the wood lemming (*Myopus schisticolor*), which inhabits exclusively coniferous forest with a thick moss cover (Eskelinen, 2002; Calandra et al., 2015). Such links between vegetation and rodents, imply that all abiotic (e.g. climate, CO<sub>2</sub> concentration)

and biotic (e.g. pest distribution, other consumers) parameters that affect vegetation subsequently impact rodents. Since the Bioclimatic Analysis is based only on correlations, and not causations, between faunal components and climate variables, all these interrelations are indirectly taken into account.

# 3.2 Modern climate

#### 3.2.1 Climate zone classification

The climatic classifications obtained using modern rodent associations almost perfectly capture the distribution of climatic regions of the western Palearctic (Fig. 2). Only 15.9 % (n=25) of localities were not well predicted, which is in agreement with the apparent error (13 %) published by Royer et al. (2020). Nevertheless, most wrong predictions tend to be concentrated around ecotonal areas between different climate zones (Fig. 2), particularly in the transition regions between the temperate area (VI) and the boreal climate zone (VIII) in southern Scandinavia or between the desert (III) and the Mediterranean climate (IV) along the northern African coast and Middle-East. Few of these localities show posterior probability values below 0.95, suggesting that the model has difficulties in distinguishing the mixed faunal associations between two climatic zones. A number of classification errors comes from regions with ample climatic variations and highly diverse environments at local and regional scales, such as the Anatolian and Iranian plateaus. They include mountainous areas home to a very wide range of rodents with distinct ecological affinities, from deserts and semi-deserts (Rhombomys, Meriones) to steppes (Ellobius, Cricetulus) and woodlands (Apodemus), which leads to mixed rodent communities. The classification posterior probabilities do not systematically detect these mix associations, although some were found in northern Anatolia and in the Balkans. Similarly, mountainous regions may also be associated with classification errors (e.g. the Apennines), due to the mixture of lowland species (several species easily survive up to 2,000 m) and the relatively few species typical of high mountain environments (*Marmota*). Finally, some errors may be associated with insularity, where faunal associations are the result of the process of Holocene island isolation and the introduction of species by humans. For instance, in Ireland the absence of many vole species and the only presence of wood mouse and squirrel represent a clear case of out-of-equilibrium communities that act to bias climate classifications. Nonetheless, despite all these potential sources of errors (communities from ecotonal, highly heterogeneous or areas with peculiar biogeography history), the climate of 84 % of the modern rodent assemblages was correctly classified.

Figure 2: Present-day climatic classifications obtained using modern rodent associations. Colors represent the climatic zone estimated by the Bioclimatic Model (see Table 1). Triangles represent well-classified localities, while squares depicts not well-classified localities in comparison with their modern situation (see Table C1). These last symbols show two sizes, the larger one representing the well-established localities by Bioclimatic Model with a posterior probability values higher than 0.95 while the small one presents localities with a posterior probability values obtained smaller than 0.95 by the Bioclimatic Model.

# 3.2.2 Temperature interpolations

Our rodent-based reconstructions of modern MAT, MTWA and MTCO depict a geographical distribution that is consistent with the spatial pattern shown by GCMs, Beyer2020 and ERA5 (Fig. 3). The area-

averaged temperature values estimated by rodent-based reconstructions reach 11.2 °C for the MAT, 23.6 °C for the MTWA and -0.4 °C for the MTCO (Table 2, Fig. 4).

Figure 4: Bar plot of area-averaged MAT, MTWA and MTCO for present-day and the LGM, as estimated from rodent associations, ERA5 and multiple GCMs. The area average is based on the 55 locations not affected by maximum ice-sheet extension (see text and Table 2).

| Model                                                  | Period              | MAT   | MTWA  | мтсо  | MAT anomalies | MTWA anomalies | MTCO anomalies |
|--------------------------------------------------------|---------------------|-------|-------|-------|---------------|----------------|----------------|
| Interpolations<br>from rodent<br>association<br>models | Present-day         | 11.18 | 23.57 | -0.36 |               |                |                |
|                                                        | YD                  | 8.65  | 23.06 | -4.67 | -2.53         | -0.51          | -4.31          |
|                                                        | Allerød             | 8.60  | 22.12 | -3.90 | -2.58         | -1.45          | -3.54          |
|                                                        | Bølling             | 7.58  | 22.35 | -7.62 | -3.60         | -1.23          | -7.26          |
|                                                        | Heinrich<br>Stadial | 8.08  | 22.22 | -5.29 | -3.10         | -1.35          | -4.93          |
|                                                        | LGM                 | 5.91  | 20.21 | -7.16 | -5.27         | -3.36          | -6.80          |
|                                                        | Present-day         | 11.47 | 20.66 | 2.51  |               |                | _              |
|                                                        | YD                  | 9.76  | 22.32 | 0.55  | -1.71         | 1.66           | -1.96          |
|                                                        | Allerød             | 9.63  | 21.76 | 0.51  | -1.85         | 1.10           | -2.00          |
| Beyer2020                                              | Bølling             | 8.06  | 18.70 | -0.75 | -3.41         | -1.97          | -3.26          |
|                                                        | Heinrich<br>Stadial | 6.98  | 17.02 | -2.20 | -4.49         | -3.64          | -4.71          |
|                                                        | LGM                 | 5.99  | 14.81 | -2.74 | -5.49         | -5.85          | -5.25          |
| ERA5                                                   | Present-day         | 12.28 | 20.02 | 3.89  |               |                |                |
| CCSM4                                                  | Present-day         | 13.48 | 22.97 | 6.09  |               |                |                |
|                                                        | LGM                 | 7.44  | 16.33 | -0.58 | -6.04         | -6.64          | -6.67          |
| CNRM-CM5                                               | Present-day         | 11.07 | 21.72 | 2.58  |               |                |                |
|                                                        | LGM                 | 8.15  | 21.58 | -1.98 | -2.92         | -0.14          | -4.56          |
| IPSL-CM5A-LR                                           | Present-day         | 11.09 | 21.46 | 2.59  |               |                |                |
|                                                        | LGM                 | 3.13  | 14.24 | -6.55 | -7.96         | -7.22          | -9.14          |
| MIROC-ESM                                              | Present-day         | 13.81 | 23.16 | 6.46  |               |                |                |
|                                                        | LGM                 | 7.21  | 15.77 | -0.51 | -6.60         | -7.39          | -6.97          |
| MPI-ESM-P                                              | Present-day         | 12.79 | 20.87 | 4.59  |               |                |                |
|                                                        | LGM                 | 6.48  | 14.73 | -2.31 | -6.31         | -6.14          | -6.90          |
| MRI-CGCM3                                              | Present-day         | 11.56 | 20.76 | 3.81  |               |                |                |
|                                                        | LGM                 | 6.67  | 17.69 | -1.60 | -4.89         | -3.07          | -5.41          |
| COSMOS-ASO                                             | LGM                 | 5.41  | 16.08 | -4.43 |               |                |                |
| FGOALS-g2                                              | LGM                 | 4.35  | 15.75 | -5.66 |               |                |                |

Table 2: Area-averaged MAT, MTWA and MTCO in rodent-based estimations, ERA5 and PMIP4 and Beyer2020 GCMs for different time periods covered by the data (present-day, LGM, Heinrich Stadial, Bølling, Allerød and Younger Dryas). The area average includes the 55 locations (see text and Table 2). Anomalies against the present-day conditions are also shown.

The averaged MAT value is 1 °C lower than the ERA5 value and is very close to 4 of the 6 GCMs values (Table 2, Fig. 4). At the site scale, despite the presence of few obvious outliers, the MAT rodent-based reconstruction values are within the GCMs uncertainty range (Fig. 5A). The most sizable pattern for MAT is the abrupt decrease in temperature between climate zones VI and VIII, a pattern captured by our rodent-based reconstruction despite cold biases in the climate zone VIII. The relationship between rodent estimations and ERA5 is strong ( $R^2 = 0.92$ ), comparable with the relationship between ERA5 and the GCMs (with R² ranging between 0.87 and 0.97: Fig. 5B). MAT values obtained from Bioclimatic Analysis of rodent assemblages show spatial discrepancies with ERA5 (Fig. 6). There is a tendency towards cold biases (~-3/-4 °C) all along the Atlantic coast latitudinal gradient, contrasting with positive biases (up to ~+3 °C) over Ukrainian and Russian steppes (Fig. 6). Similar bias patterns are observed when comparing ERA5 with GCMs and Beyer2020 estimations, but with weaker differences, as in Iberian Peninsula or in Scandinavia (Fig. 6).

Figure 5: A) Present-day area-averaged MAT estimated from 157 rodent associations. Triangles represent localities whose modern climate zone was correctly classified, while squares are localities wrongly classified. Colors represent the climatic zone estimated by the Bioclimatic Model (see Table 1). MAT estimations from Beyer et al. (2020) (grey circles) and ERA5 values (black circles) are also shown. The boxplots represent MAT estimations from six different PMIP4 GCMs. Localities were ordered according to modern MAT values from ERA5 estimations. B) Scatterplots and linear regressions between present-day MAT from ERA5 and the estimations from rodent assemblages or from the GCMs. All of them are statistically significant (p < 0.05) and R<sup>2</sup> is shown.

Figure 6: Differences in MAT, MTWA and MTCO between rodent-based, PMIP4 GCM ensemble mean and Beyer2020 GCM and the ERA5 reanalysis. Warm colors indicate that the models estimate higher temperatures than ERA5, while cold colors indicate the opposite.

Compared to ERA5, MTWA values are overestimated by all models over Europe. The warm bias reaches almost 4 °C for rodent-based reconstructions, 2 to 3 °C for GCMs and less than 1 °C for Beyer2020 (Table 2, Fig. 4). At the site scale, although most rodent-based reconstruction values fall within the range of the

GCM uncertainty, they are usually in the warmer range of their estimations (Fig. 7A). Rodent associations generally infer warmer conditions than ERA5 and Beyer2020 for most of the climate zones, particularly for climate zones VI and VIII (Fig. 6 and 7A). Climate zones III and VII, on the other hand, show more accurate estimations (Fig. 7A). The regressions between ERA5 and the rodent estimates remain, however, still satisfactory (R<sup>2</sup> = 0.84) as GCMs (R<sup>2</sup> = 0.79-0.93), which indicates that the rodent-based model captures relatively well the spatial pattern of MTWA variation, although the slope and intercept going significantly away from 1 and 0 respectively (Fig. 7B). Discrepancies between rodent-based estimations and ERA5 show a similar pattern to GCMs biases, although somewhat more pronounced. It generally consists of higher temperature estimates, especially at latitudes above 50°N and lower estimation under 40°N (Fig. 6).

Figure 7: A) Present-day area-averaged MTWA estimated from 157 rodent associations. Triangles represent localities whose modern climate zone was correctly classified, while squares are localities wrongly classified. Colors represent the climatic zone estimated by the Bioclimatic Model (see Table 1). MTWA estimations from Beyer et al. (2020) (grey circles) and ERA5 values (black circles) are also shown. The boxplots represent MTWA estimations from six different PMIP4 GCMs. Localities were ordered according to modern MAT values from ERA5 estimations. B) Scatterplots and linear regressions between present-day MTWA from ERA5 and the estimations from rodent assemblages or from the GCMs. All of them are statistically significant (p < 0.05) and R<sup>2</sup> is shown.

The averaged MTCO value estimated from rodent associations is 4 °C lower than ERA5, and 2.8-6.7 °C lower than the GCMs (Table 2, Fig. 4). At the site scale, rodent-based estimations are also usually colder than the ERA5 and GCMs values, in particular for sites located in climate zone VIII (Fig. 8A). The

common variance between ERA5 and rodent-based estimations is weaker ( $R^2 = 0.78$ ) than with the GCMs ( $R^2 = 0.85$ -0.96). This is mainly due to large discrepancies in site located within temperate (VI) and boreal (VIII) climate zones (Fig. 8B). Spatially, southern locations are skewed toward warmer temperature inferences, without bias exceeding 5 °C. On the other hand, the Atlantic, Baltic and Arctic facades offer colder inferences (Fig. 6).

Figure 8: A) Present-day area-averaged MTCO estimated from 157 rodent associations. Triangles represent localities whose modern climate zone was correctly classified, while squares are localities wrongly classified. Colors represent the climatic zone estimated by

the Bioclimatic Model (see Table 1). MTCO estimations from Beyer et al. (2020) (grey circles) and ERA5 values (black circles) are also shown. The boxplots represent MTCO estimations from six different PMIP4 GCMs. Localities were ordered according to modern MAT values from ERA5 estimations. B) Scatterplots and linear regressions between present-day MTCO from ERA5 and the estimations from rodent assemblages or from the GCMs. All of them are statistically significant (p 

Figure 9: A) Rodent-based estimations of MAT, MTWA and MTCO for the LGM period. B) Difference between the rodent anomaly values calculated between the LGM and the present-day periods (e.g. LGM minus present-day), and those calculated for Beyer2020 and from GCMs models. Warm colors indicate that rodent-based estimations provide higher temperature changes than the PMIP4 or Beyer2020 GCM estimations, while cold colors indicate the opposite. Colored triangles represent the climatic zone estimated by the fossil rodent associations using the Bioclimatic Model for the LGM period (Table 1). These symbols show two sizes, the larger one representing the posterior probability values higher than 0.95 while the small one presents the posterior probability values smaller than 0.95.

The temperatures estimated from the LGM rodent associations show a strong latitudinal gradient (Fig. 9A), with an average MAT anomaly of -5.3 °C compared with present-day temperature (Table 2). This is consistent with Beyer2020 (-5.5 °C) and within the range of uncertainty of PMIP4 GCMs (-3 °C to -8 °C). Rodent estimations from southwestern France range from 2 to 5 °C and are relatively similar to

estimations from Beyer2020 (6 °C) and PMIP4 GCMs (3 °C to 11 °C). Rodent estimations reach around -3 °C at Brussels, while those from Beyer2020 are around 2.5 °C and PMIP4 GCMs range from -4 °C to 5 °C (Fig. B2). They are also 2-3 °C colder than other estimations based on faunal species (Puzachenko & Markova, 2023). Our rodent-based model suggests that the largest changes in MAT between LGM and present-day conditions are observed in the Russian polar latitudes and around the British-Irish ice sheet (anomalies between -5 and -10 °C), while the cooling at lower latitudes was comparatively attenuated (anomalies between 0 and -5 °C) (Fig. 10), as also observed from pollen-based climate reconstruction (e.g. Bartlein et al., 2011; Davis et al., 2022). Spatial patterns from rodent-based model exhibit thus a stronger latitudinal gradient during the LGM than today, and differ with Beyer2020 and PMIP4 GCMs (Fig. 9B & 10; Fig. B2), the latter proposing a less marked latitudinal gradient in MAT across the western Palearctic. This leads rodent-based model to propose that MAT anomalies between LGM and present day conditions were higher than PMIP4 GCMs and Beyer2020 in northwestern part of Europe (in blue in Fig. 9B) and smaller in the southwestern and eastern part of Europe (in red in Fig. 9B). Particularly notable differences in anomalies are observed in the eastern area for which Beyer2020 models suggest larger MAT changes than rodents, as in southern Ural, where rodent anomaly estimations are small; this is also probably emphasized due to the absence of rodent associations in the southeastern part of the study area for this time interval, which may produce spurious temperature inferences for lower southeastern areas (see red area in Fig. 10).

Figure 10: MAT, MTWA and MTCO anomalies calculated between present-day and LGM conditions (LGM minus present-day) for the rodent-based, PMIP4 and Beyer2020 estimations. Cold colors indicate lower temperatures during the LGM compared to present-day, while warm colors signify the opposite.

Rodent-based changes in MTWA and MTCO between present-day and LGM conditions are similar to those obtained for MAT, although changes in MTCO are stronger than MTWA (Table 2, Fig. 10 & Fig. B2). On average the change in MTCO reaches almost -7 °C, while the MTWA anomalies only reach -3.4

°C (Table 2), with almost lowest LGM to present-day changes for the southern latitudes. It is in agreement with Davis et al. (2022), who show that LGM climatic cooling was significantly greater in winter than in summer using updated LGM pollen estimations in Eurasia (Peyron et al., 1998; Jost et al., 2005; Wu et al., 2007). Our estimations are relatively close to Beyer2020 and PMIP4 GCMs (Table 2). However, spatial patterns differ significantly between the rodent estimations and the other models. Rodent-based estimations of MAT, MTWA and MTCO are colder in the northwest region and warmer in the eastern region compared to Beyer2020 and PMIP4 GCMs (Fig. 9B).

Figure 10 summarizes temperature changes between present-day and LGM conditions across all models and indicators. A pronounced meridional gradient is evident in all models and for all indicators, with stronger cooling in the north than in the south. In addition to this meridional gradient, the rodent model reveals a geographical gradient, showing LGM temperatures that are much colder in the northwest and slightly warmer in the southeast compared to present-day conditions. This pattern is consistent across MAT, MTWA, and MTCO, but does not appear in Beyer2020 or the GCMs. It is linked to the presence of an Arctic rodent association in the northwest, possibly indicating lower heat input associated with a weak AMOC. The lack of this zonal gradient in GCMs may be due to an overly strong AMOC, despite improvements from PMIP3 to PMIP4 (Kageyama et al., 2020).

As noted by Davis et al. (2022), although the traditional view of the LGM has been defined for many years, many questions still remain to be explored, when more precise quantification is researched. The LGM is depicted as a very different period compared to today, with distinct climate, vegetation and landscape, together with a different orographic context including large ice-sheets, which have fully

impacted faunal communities. Typical cold rodent species have been recovered throughout mid- and high latitudes, and were mainly limited by natural barrier like mountains to reach southern latitudes. They form ecotone rodent communities, mixing species from polar, boreal and temperate environments, and are far from reflecting typical full periglacial communities, except for the ones located at the closest range of the ice-sheet. The quantification obtained with the rodent-based model do support the global view of a colder LGM world, but with MAT anomalies no larger than -5.5 °C compared to the present-day conditions, and underpins that changes in MTCO were stronger than in MTWA. Rodent associations produce nonetheless spatial temperature pattern different than the ones estimated from Beyer2020, suggesting notably greater temperature changes in the northwestern part of Europe, while the temperature anomalies in the southwestern and the eastern part of Europe were weaker than Beyer2020 and PMIP4 GCMs.

# 615 3.3.2 Once upon a time in the Late Glacial

Compared to the LGM picture, the Heinrich Stadial experienced sizable changes in species distribution throughout the western Palearctic (e.g. Royer, 2016; Royer et al., 2016; Markova et al., 2019). However, climate zone distribution remains globally similar between these two periods (compare Fig. 11 with Fig. 9A). Northern Europe yielded more sites with Arctic rodent associations, like in Belgium, Poland and the northern Ural Mountains (Teterina, 2009; Stewart & Parfitt, 2011; Ponomarev et al., 2012; López-García et al., 2024), forming a composite mix of boreal (VIII) and polar (IX) climatic zones. The transition between climatic zones VI and VIII appears south of 50° N in the western part of Europe and around the 55° N in the eastern part, while the central Europe remain dominated by an arid-temperate climate zone

(VII), as shown in Fig. 11. The Iberian and Italian peninsulas show both a mix of Mediterranean (IV) and temperate (VI) climate zones. With the warming of the Bølling and the Allerød, large changes occurred at species level (Kalthoff, 1998; Jeannet, 2009; Royer et al., 2016, 2021; Rofes et al., 2020; Wong et al., 2017). Changes in the climate zone distribution occur slightly with boreal climate zone (VIII) in western Europe retreating towards northern latitudes, as the ice sheet melted. The temperate climate zone (VI) slightly develops to the north up to 50°N, while the central and eastern Europe seems to present a pattern relatively similar to previous intervals with the presence of an arid-temperate climate zone (Fig. 11). Yet, to date, there is still a general lack of information on the southeastern part of the study area for this time interval making it difficult to understand how exactly the arid-temperate climate zone (VII) has changed. The Iberian and Italian Peninsulas still deliver rodent associations of both Mediterranean (IV) and temperate (VI) climate zones, although the latter are becoming rare. There are few fossil sites to cover the Younger Dryas and most of them are in Western Europe. They do not reveal remarkable changes in climate zone distribution, despite the cooling of this event. The British Isles still experienced boreal (VIII) and polar (IX) climate zones attested by the presence of typical boreal and polar lemming (Price, 2003; Arbez et al., 2021), while a boreal climate zone sets up in the northern part of Urals (Fig. 11).

Figure 11: Rodent-based estimations in MAT, MTWA and MTCO for the Heinrich Stadial, Bølling, Allerød and Younger Dryas periods. Colored triangles represent the climatic zone estimated by the fossil rodent associations using the Bioclimatic Model (Table 1). These symbols show two sizes, the larger one representing the posterior probability values higher than 0.95 while the small one presents the posterior probability values smaller than 0.95.

Average climate estimations obtained from rodent associations present a gradual temperature warming between LGM and present day (Table 2). MAT of different Late Glacial events remains however within a relatively narrow range of values (7.6 to 8.7 °C), with MAT value at 8.1 °C since the Heinrich. MTWA values vary little (between 22.1 and 23.1 °C), while MTCO values range from -7.6 to -3.9 °C. These estimations obtained with rodent-based model are relatively close to those of Beyer2020, in particular for MAT (Table 2). For MTWA and MTCO, Beyer2020 estimations suggest larger temperature changes across the events, while rodent-based reconstructions suggest only small changes in MTWA and still larger MTCO anomalies for Bølling-Allerød-Younger Dryas periods (Table 2).

These spatial temperature reconstructions of Late Glacial events differ from Beyer2020 in the same way (Fig. 12). Except the southeastern part for which no rodent association is available, the main disagreements come from a different latitudinal gradient of changes in the western part of Europe, and differences in northeastern part of Europe. For the northwestern part of Europe, the rodent-based model estimates that MAT, MTWA and MTCO anomalies between Late Glacial events and present day conditions were systematically higher than Beyer2020 (in blue in Fig. 12), suggesting thus larger temperature anomalies throughout all the Late Glacial. For instance, for the Bølling and the Allerød, MAT anomalies from rodent-based model are estimated around -12.5 / -8 °C in England, compared with -5 / -3.5 °C for the Beyer2020 model. By contrast, the rodent-based model yields smaller temperature anomalies for the southwestern part of Europe, as well as for the eastern part of Europe during the Heinrich Stadial and Bølling. For instance, MAT anomalies of Heinrich Stadial and Bølling in the southwestern France are estimated around -3 / -3.5 °C for the rodent, thus relatively close of the one of Beyer 2020 estimated at -4.5 / -5 °C. For the northeastern part of Europe, Beyer 2020 estimated greater

anomalies for MAT and MTCO between Late Glacial events and present day (in red in Fig. 12) than the rodents. For the region of Moscow, MAT anomalies of Beyer2020 range from -13.5 to -11 °C for the Heinrich Stadial and Bølling, while rodent-based model estimated MAT anomalies between -9 and -10 °C.

Figure 12: Difference between the rodent anomaly values calculated between the past (Heinrich Stadial, Bølling, Allerød and Younger Dryas) and the present-day periods (e.g. LGM minus present-day), and those calculated for Beyer2020 models. Warm colors indicate that rodent-based estimations provide higher temperature changes than Beyer2020 estimations, while cold colors indicate the opposite. Colored triangles represent the climatic zone estimated by the fossil rodent associations using the Bioclimatic Model for the past period (Table 1). These symbols show two sizes, the larger one representing the posterior probability values higher than 0.95 while the small one presents the posterior probability values smaller than 0.95.

As a consequence, temperatures obtained from rodent-based reconstructions are far from changing uniformly across Europe between the Late Glacial events (Fig. 13). It shows larger spatial variations than Beyer 2020 temperature changes. Indeed the Beyer 2020 model shows more spatially uniform changes for the MAT across Europe between the Late Glacial events, with main changes located in areas close to the ice-sheet, while some longitudinal variations have been observed for MTWA and MTCO estimations (Fig. B3). Conversely, rodent-based model suggests greater latitudinal and longitudinal variation in changes. After the LGM, the increase of MAT rodent-based estimations during Heinrich Stadial is larger along the Atlantic seaboard than in central Europe. Southwestern France and the north of Spain experienced a warming between 1.5 to 3 °C, while temperature changes are lower than 1 °C in Belgium, Germany or Poland. The northern eastern part shows an increase of 3 °C and 6 °C for MAT and MTCO, respectively. Beyer2020 suggests a general raise of around 2 °C for MAT, with increase no larger than 1.5 °C in southwestern France and Spain, and between 1.2 and 5 °C in Belgium, Germany or Poland. During the Bølling, MAT and MTCO fall with the rodent-based model for the middle and the lower latitudes, while they rise in the northwest, as in Belgium and Germany, where a MAT increase of 2 to 3 °C is found. The Allerød presents sizable changes, with a MAT warming between 1 and 3 °C in all western Europe. The southern latitude is mainly impacted by an increase in MTCO, with for instance increase of 3.5 °C for the north of Spain and of 2 °C for southwestern France. The northern latitudes show an increase in MTWA, reaching up to 2.5 °C as in Brussels. The Younger Dryas event brings back well-marked glacial conditions before the beginning of the current interglacial, particularly along the Atlantic coast and the British Isles. The latter experienced MAT decrease exceeding 5 °C (Fig. 13), while Beyer2020

shows no sizeable changes in this areas, except in Scotland, with an MAT increase around  $2.5\,^{\circ}$ C (Fig. B3).

Figure 13: Differences in MAT, MTWA and MTCO between each successive time periods in rodent-based estimations. Triangles and circles represent the position of fossil localities used in each pair of time intervals. The ice sheet of the most recent period is represented in blue, while the limits of the older one appears in grey.

The Late Glacial period experienced significant climatic shifts, marked by rapid climate oscillations, which led to a dynamic re-shaping of living spaces, with the progressive retreat of ice sheets freeing up new areas in the northern regions, while rising sea levels simultaneously submerged substantial portions of coastal regions. These changes have restrained other areas, isolated some islands and induced significant changes in plant communities. All these biotic and abiotic changes have led to considerable changes in faunal associations throughout Europe, triggering range shifts, local extirpations, extinctions, population replacement, isolation in refugia or recolonization processes (e.g. Stewart et al., 2010; Brace et al., 2016; Baca et al., 2017; Puzachenko & Markova, 2019; Magyari et al., 2022). At the community scale, the changes remain slow as shown by the progressive rise of temperate climate zone (VI) throughout the Late Glacial (Fig. 11), and in the early Holocene for the northern latitude. In southwestern France, the changes in small mammal associations are gradual taking all of the Late Glacial events to switch progressively from the typical glacial community to the typical interglacial community (Royer et al., 2016), including also the presence of certain species only at certain events (Royer, 2016; Royer et al., 2018). These Late Glacial periods produce therefore rodent associations mixing typical taxa of northern environments, with new southern and eastern taxa (Royer et al., 2016), which gradually evolve in latitude in line with the events (e.g. Cordy, 1991; Wong et al., 2020; López-García et al., 2024). These faunal changes reflect temperature changes that vary both in time and space, producing spatial pattern that differs from Beyer2020. Rodent results suggest that the northwestern part of Europe experiences larger

temperature changes, while temperature anomalies in the southwestern and the eastern part of Europe were smaller than those estimated by Beyer2020.

#### 3.3.3 How to interpret the differences between GCMs and rodent-based inferences

As described previously, rodent-based model provides patterns similar to GCMs for both present-day and past conditions, with for instance a basic picture of LGM as a cold world with MAT anomalies around - 5.5 °C in the Western Palearctic. Yet, there are notable divergences among the approaches. These divergences are generally included in the range of GCM uncertainties and hide sizable regional differences, like between northwestern and northeastern Europe (Fig. 9B & 10). Compared to GCMs, rodent associations of LGM and Late Glacial periods generate colder temperature values in the western part of Europe and warmer in eastern Europe (Fig. 9B). This leads to a stronger west-east temperature gradient in the rodent-based model against both PMIP4 GCMs and Beyer2020, especially for the coldest months.

The physical interpretation of these distinct temperatures observed for past period is questionable. On the one hand, working with faunal remains require to take into account few taphonomical, analytical and ecological assumptions detailed in section 3.1. The discrepancies between our rodent-based model and PMIP4 GCMs and Beyer2020 in higher latitudes (Fig. 10) could be explained by a limitation in the Bioclimatic Analysis. Therein, MAT of the coldest areas used to generate the new version of the Bioclimatic Analysis reach around -15 °C (Royer et al., 2020), in areas with typical arctic species as lemmings or tundra shrews. It is then very difficult to properly extrapolate MAT values below -15 °C,

while some GCMs estimations easily reach up to -30 °C and even colder in eastern regions, questioning how faunal species identified through the fossil record could have survived in such conditions.

On the other hand, GCMs have their own uncertainties, biases, and sensitivity to changes in the radiative forcing and boundary conditions. Each GCM uses different sets of parameterizations to approximate subgrid processes that cannot be resolved at coarse spatial and temporal resolutions, which strongly influence climate simulations (Taylor et al., 2012; Varela et al., 2015), resulting in a large range uncertainties across GCMs. GCMs often diverge from actual paleontological data (Jost et al., 2005; Allen et al., 2008; Braconnot et al., 2012; Latombe et al., 2018). Beyer2020 model appears to propose the best estimation for modern context (Fig. 4, 5, 7 & 8), due to bias correction. Interestingly, they evaluated their method for past period at the global scale, comparing it only to mid-Holocene and LGM pollen database (Bartlein et al., 2011). The question may arise regarding the validation of this model during unstable glacial periods when relying solely on a single proxy. This has been recently criticized, because some of the pollen sequence were not necessarily properly well attributed to the LGM (Davis et al., 2022). Furthermore, LGM is a peculiar period of the last glaciation, with a specific abiotic context. No validation has been tested for the others glacial periods, which present their own abiotic particularities.

On the basis of all these assumptions, we can finally conclude that these divergences between GCMs and rodent-based estimations could result from multiple factors that need further investigations. Beyond GCM limitations, our interpolations are "a point of view" of rodent associations, which could contribute to explain the disagreements with GCMs. Species association may exist under environmental conditions that do not exist today, but did exist in the past (Jackson & Overpeck, 2000; Williams & Jackson, 2007;

Lyman, 2017). The relationships between the species and their environment are in constant evolution, and the current observations reflect only a realized niche, potentially underestimating the plasticity and adaptive capacity of species in response to new abiotic and biotic constraints. Rodent-based estimations tend to show a slightly different zonal temperature gradient from the GCMs, with colder values in western Europe and warmer in eastern Europe (Fig. 9, 10 & 12). This gradient may be slightly amplified by the lack of spatial data in some regions. In the present case, some errors can be explained by the absence of close fossil sites (e.g. near the Scandinavian ice sheet), or their limited number (as in British Isles). Our rodent estimations give MAT values in the eastern Europe that are warmer than those provided by GCMs for all the time intervals of the Last Glacial period. These differences could also be related to differences in faunal communities and their paleoecology. The glacial world of the LGM and Late Glacial could not be simply compared to the current interglacial biomes. The very nature of these particular environments is not comparable, due to their unique abiotic parameters like diminished the solar radiation and the CO<sub>2</sub> atmospheric concentrations, and biotic factors as the important presence of a thriving mammalian megafauna. This megafauna actively shaped European vegetation structure (Malhi et al., 2016; Pearce et al., 2023; González-Varo, in press), potentially driving the diversification of distinctive rodent communities.

#### 4 Conclusion

Rodent associations are a regularly used proxy to reconstruct local past climate and environment contexts, as they are often found in archaeological and paleontological excavations and are highly sensitive to

environment changes. Here, we focused our study on large spatial scales, over the western Palearctic, to access continental-scale paleoclimate reconstructions based on a total of 279 archaeological and paleontological levels. Although there are entire regions with few or no data, our data set encompasses a wide range of latitudinal gradients, providing distinct climatic contexts, which give us the opportunity to address the spatial background for changes in past climate conditions across the region.

The rodent-based climate reconstructions allow for exploring spatial changes between present day conditions and different time periods ranging from the Last Glacial Maximum (LGM) to the end of the Late Glacial (i.e., 23,000 to 11,700 years before present). Our main findings highlight that rodent-based reconstructions:

- are efficient for examining large spatiotemporal variations of past climate, enabling detailed investigations of temperature changes at continental scale, since they accurately capture the modern distribution of climatic zones in the Western Palearctic, as well as (ERA5) spatial patterns of macroclimate, notably for MAT;
- exhibit a stronger latitudinal gradient during the LGM than today, with an average MAT anomaly of 5.3 °C relative to present-day conditions and more pronounced changes in MTCO than in MTWA;
  - suggest a gradual warming on average from the LGM to the Late Glacial. Nonetheless, the spatial pattern of temperature change between these two periods was highly heterogeneous across Europe, with a faster

increase of temperature in the northwestern parts of the continent than in the southwestern and eastern regions;

- produce spatial temperature patterns that are more different than those derived from GCMs, although these divergences are generally included in the range of GCM uncertainties, which points out the relevance of using paleoclimatic approaches based on multiple proxies.

Future work in under-sampled regions such as the Balkans, North Africa or the Middle East, is expected to enhance the robustness of our conclusions. This could allow for the integration of rodent-based inferences as input or comparison data for GCMs, and to new comprehensive analyses including other additional proxies. A similar approach including other taxa such as Eulipotyphla or large mammals could also be developed to explore, in particular, other climatic variables such as precipitation or aridity. These initiatives should further improve our knowledge of past climate changes.

### 815 Acknowledgements

This research was partially financed by PID2022-138275NB-I00 projects (Ministry of Science and Innovation, Spain), S.G. was funded by the Ministry of Universities and the Next Generation European Union programme through a Margarita Salas Grant from Universidad Complutense de Madrid (CT31/21). I.M. was funded by the Alexander von Humboldt Foundation through a Humboldt Postdoctoral

Fellowship.

Data and code availability

All the analyses of the paper were performed using the R software (R core team, 2023).

The Bioclimatic Analysis is freely available through the PalBER R scripts (available at

https://github.com/AurelienRoyer/PalBER). All the R scripts generated for all calculations, as well as the

modified functions, and data are available at https://github.com/AurelienRoyer/Climate-spatial-

Interpolations-LGM-LG, DOI: 10.5281/zenodo.14905209, in order to be fully reproducible.

**Supplement** 

https://github.com/AurelienRoyer/PalBER

https://github.com/AurelienRoyer/Climate-spatial-Interpolations-LGM-LG,

DOI: 10.5281/zenodo.14905209

**Author contributions** 

A.R. designed the study. B.L., I.M. and M.H.F. provided the data on the modern rodent assemblages and

Climatic Restriction Index of rodent species. J.C. and B.P. provided GCMs and ERA5 data. A.R. provided

rodent fossil data. A.R., R.L. and J.C. developed the methodology. A.R. wrote R scripts. A.R., J.C., R.L.,

S.B., B.L., I.M., B.P., S.M. M.H.F. participated in analyzing and writing, provided edits and comments

on subsequent drafts.

### **Competing interests**

The contact author has declared that none of the authors has any competing interests.

## **Financial support**

This research was partially financed by PID2022-138275NB-I00 projects (Ministry of Science and Innovation, Spain), S.G. was funded by the Ministry of Universities and the Next Generation European Union programme through a Margarita Salas Grant from Universidad Complutense de Madrid (CT31/21).

I.M. was funded by the Alexander von Humboldt Foundation through a Humboldt Postdoctoral Fellowship. A.R was financed by the Biogéosciences laboratory resources.

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
