# Peer review of "Late Pleistocene temperature patterns in the Western Palearctic: insights from rodent associations compared with General Circulation Models"

_EGUsphere, 2025_

## Author Comment (AC1)

**We would kindly thank Reviewer 1 for taking the time and consideration to review our manuscript, which helped to improve its clarity. Below we address each individual comment in Blue.**

Dear authors,

The manuscript is very interesting and with the R script to reproduce de data very useful for the for researchers dedicated to the study of climate and landscape reconstruction in the quaternary small mammals. However, I consider that the manuscript could be published after clarifying some doubts and correcting some minor issues:

Abstract: Abbreviation of General Circulation Model is wrong written is GCMs

**We fixed this spelling error**

Keywords: In my opinion there are two unnecessary ones (paleoclimate and Mammalia) and they are not well ordered: rodent-based reconstructions, temperature patterns, Last Glacial Maximum, Late Glacial, Heinrich Stadial, Bølling-Allerød Interstadial, Younger Dryas.

**We corrected the keywords as proposed by the reviewer**

page 3, line 72: (Rodentia and /or Eulipotyphla)

**We have modified as proposed**

page 9, line 172-173: write in brackets the scientific name of the species after de common name, *Dicrostonyx torquatus* and *Apodemus agrarius*

**We added the scientific names of these rodents**

page 7-lines 141-142. Please better explain the use of the IUCN Red List for species distribution per 50 km, because those maps do not appear to be in the IUCN, 2021.

**We clarified in the text that we used the geographic range polygons available from the IUCN Red List spatial data, and performed spatial intersections in R to determine whether each species range overlaps with a 50 km radius circle around each studied locality. This allowed us to generate locality-specific rodent faunal lists. We have revised the Methods section accordingly to make this process clearer:**
**"For each locality, we compiled a rodent faunal list (Table C2) by identifying all rodent species whose geographic range polygons, sourced from the IUCN Red List spatial data (IUCN, 2021), intersect with a 50 km radius buffer around the locality coordinates."**
**We hope this clarification resolves your concern.**
**We also added in supplementary material, the R script allowing to reproduce this part of data treatment.**

page 10-line 176: Why you don't use direct the updated version of OxCal 4.4.4. (Bronk Ramsey, 2021) and the IntCal20 (Reimer et al. 2020). Probably some of the dating that you calibrated are wrong without the updating. Check it.

**The main reason for not using the updated version of OxCal is purely a question of timing with the project; the first version of the database having been built before the release of IntCal20. We have not updated because, although the latest version is clearly better, the changes in the latest calibration mainly concern the Holocene and MIS3; the changes for the LGM and Late Glacial periods remain relatively minor.**

**We have recalibrated all dates with the updated versions of IntCal. No major changes have been observed with these latest versions. If we quantify the changes related to the used of the new curve, we have an average change in the minimum and maximum bounds of the intervals of just over 51 years, and only 9 levels undergo changes between 200 and 500 years (generally the aging ones).**

**We have modified the text accordingly with the new references. We modified the table C2 with the new dates.**

page 13-238-241. In my opinion, the reason why altitude is omitted in the article needs a better explanation. Although the sites used are few, the altitudinal location is very important. At the same longitude and latitude, at different altitudes, you can have completely different associations of micromammals.

**We agree with the reviewer that altitude is a very important factor for climate, ecosystems and small mammals. Initially, we explored altitude by adding this factor to the latitude and longitude parameters in GLMM. However, despite the size of the database, there are not enough high-elevation sites for each period, leading to erroneous models for some periods. We therefore choose to apply an average vertical lapse rate. This compromise is not perfect by any means, as we fail to take into account the likely long-term changes in the vertical lapse rate, and the possibility that it may differ from case to case mountains (i.e. Alps, Pyrenean etc..).**

**We modified the following line to clarify the point (line 249) : "**Although the GLMM can include the effects of altitude, the number of high altitude fossil sites is too limited **per period to obtain sufficiently robust correlations to reproduce adiabatic effect, leading to erroneous models for geographical areas at middle and high altitudes. We therefore prefer to apply an average vertical lapse rate, to reflect this effect in geographical and temporal terms.** Elevation effects […]**".**

The list of sites with the different chronological periods presented is impressive and very complete. I only missed one site, published a couple of years ago, which has two levels from the end of the Late Ice Age. Here's the reference, perhaps it could be included:

Arjanto, D.Q., Fernández-García, M., López-García, J.M., Vergès, J.M., 2023. The end of Late Glacial in north-eastern Iberia: the small mammal assemblage from Cudó Cave (Mont-

Ral, Tarragona). Earth and Environmental Science Transactions of the Royal Society of Edinburgh 114 (1–2), 21–33.

**We have not selected this site and its two levels because of the extent of the period covered by the two levels 107 and 105. According to the radiocarbon dates, level 107 covers part of the LGM, but also potentially HS3, and level 105 covers several Late Glacial events. As indicated in the article, we selected "stratigraphic units associated with radiocarbon dates limited to a single time interval", in order to limit palimpsest bias and thus avoid blurring noise in the signal.**

---

## Author Comment (AC2)

**Response to Reviewer 2**

Overall, this is a strong, nice paper that infers temperature for Eurasia based on modern and fossil rodent assemblages, then compares the estimates to different model-based temperature estimates. It contributes a good perspective of faunal estimates, in contrast to the GCM and pollen-based estimates that are more common. The method that the authors use has been well-validated, and here they extend it to more sites and taxa, and do the detailed benchmarking necessary to establish rodent assemblages as a valid proxy.

*We would like to thank Reviewer 2 for their positive and very helpful review, which helped to improve the clarity and fix some mistakes. Our responses to each of the Reviewer's comments are in blue.*

My overall impression is that rodent assemblages are good proxies for MAT, less good but still strong for MTWA, but perhaps not super robust approximations of MTCO. I don't have as clear of a "takeway" for other aspects of the paper - there were interesting differences in the spatial patterns between rodents and the other estimates of paleoclimate that could indicate that the rodents do a better job than models in some places, or a worse job than models in other places. I think the lack of clear takeaways is partly because the data themselves are muddled, but it may also partly be due to the length of the discussion of anomalies and gradients, which overwhelmed the author's messages at times. I wonder if additional subheadings that state the main submessages would help with paper structure, and help draw out the main takeaways for the paper?

*We have taken your comments into account. However, rather than adding sub-sections, which we believe would not be the most effective way to highlight our results, we propose to modify the conclusion in order to clearly highlight the important results by listing them. We propose the following conclusion:*

*"Rodent associations are a regularly used proxy to reconstruct local past climate and environment contexts, as they are often found in archaeological and paleontological excavations and are highly sensitive to environment changes. Here, we focused our study on large spatial scales, over the western Palearctic, to access continental-scale paleoclimate reconstructions based on a total of 279 archaeological and paleontological levels. Although there are entire regions with few or no data, our data set encompasses a wide range of latitudinal gradients, providing distinct climatic contexts, which give us the opportunity to address the spatial background for changes in past climate conditions across the region.*
*The rodent-based climate reconstructions allow for exploring spatial changes between present day conditions and different time periods ranging from the Last Glacial Maximum (LGM) to the end of the Late Glacial (i.e., 23,000 to 11,700 years before present). Our main findings highlight that rodent-based reconstructions:*

- are efficient for examining large spatiotemporal variations of past climate, enabling detailed investigations of temperature changes at continental scale, since they accurately capture the modern distribution of climatic zones in the Western Palearctic, as well as (ERA5) spatial patterns of macroclimate, notably for MAT;

- exhibit a stronger latitudinal gradient during the LGM than today, with an average MAT anomaly of -5.3 °C relative to present-day conditions and more pronounced changes in MTCO than in MTWA;

- suggest a gradual warming on average from the LGM to the Late Glacial. Nonetheless, the spatial pattern of temperature change between these two periods was highly heterogeneous across Europe, with a faster increase of temperature in the northwestern parts of the continent than in the southwestern and eastern regions;

- produce spatial temperature patterns that are more different than those derived from GCMs, although these divergences are generally included in the range of GCM uncertainties, which points out the relevance of using paleoclimatic approaches based on multiple proxies.

Future work in under-sampled regions such as the Balkans, North Africa or the Middle East, is expected to enhance the robustness of our conclusions. This could allow for the integration of rodent-based inferences as input or comparison data for GCMs, and to new comprehensive analyses including other additional proxies. These initiatives should further improve our knowledge of past climate changes."

**Line-by-line comments:**

Line 49-50, "Our results demonstrate that rodent associations are robust proxies for reconstructing and regionalizing past climates at broad scales…". Since the authors tackle temperature only, I would narrow this to "reconstructing and regionalizing past temperature at broad scales".

We have modified as proposed.

Line 169, "obtained through adequate sample sizes". Is there a specific threshold, or taxon-specific value? Or a citation for this?

Studies on rodent fossils present variable data, especially for the older studies (list of occurrences versus NMI or number of remains). We have not applied any specific threshold. Whenever possible, we have given priority to deposits containing at least twenty individuals. Nevertheless, since this could not be applied in a systematic manner for all assemblages, we have removed that clarification in order to avoid an overstatement.

Lines 175-176, "we selected mainly stratigraphic units associated with radiocarbon dates restricted to a single time interval". This is a good step. Is there an estimate of overall uncertainty associated with site chronologies? I assume that most of the radiocarbon dates are

indirect, i.e. representing one or a few dates on a few specimens, but not all species in a unit are dated. How certain are the authors that the assemblages as a whole can be assigned to the climatic period they are associated with?

For the uncertainty associated with site chronologies, we have calibrated the radiocarbon dates obtained for each site, and we only use deposits whose calibrated date falls within the time interval of the chronozone, as indicated in the text.

As to the question of how confidently the whole assemblage can be assigned to the climatic period, we can currently only modestly answer: almost never without direct dating on rodent remains. That's why we've tried to take as much information as possible from the context. Radiocarbon dates from large mammals are a powerful indicator now regularly used to check the congruence of faunal elements (*e.g.* Costamagno et al., 2016; Mallye et al., 2018). While these dates are based on large mammals and thus provide indirect evidence, any inconsistencies in their stratigraphic context also suggest that small mammal remains are unlikely to be better preserved. Because of that, we have not included these deposits in which questionable stratigraphic context have been detected, as noted in the text. Unfortunately, such radiometric approach on small remains, offering the possibility of confirming their stratigraphic integrity, remains still particularly rare, usually limited to a single batch of small mammal remains or a single taxon (*e.g.* Woodman et al., 1997; Aguilar et al., 2008; Brace et al., 2016; Royer et al., 2018; Rofes et al., 2020; Baca et al., 2023; Ceregatti et al., 2023). We hope this should be developed more extensively in the future and systematically used, thanks in particular to the miniaturization of radiometric samples. For instance, Rofes et al. (2020) shows different cases, with some perfectly matching, as in Peyrazet site, between radiocarbon dates from large and small mammals. Radiocarbon dating is not the only indicator, and contextual information from sites derived from archaeological data, taphonomic or sedimentological contexts has also been used to determine the overall temporal integrity of each fossiliferous level.

Aguilar, J. P., Pélissié, T., Sigé, B., Michaux, J. (2008). Occurrence of the stripe field mouse lineage (Apodemus agrarius Pallas 1771; Rodentia; Mammalia) in the Late Pleistocene of southwestern France. *Comptes Rendus Palevol 7*(4), 217-225.

Baca, M., Popović, D., Agadzhanyan, A.K., Baca, K., Conard, N.J., Fewlass, H., Filek, T., Golubiński, M., Horáček, I., Knul, M.V., Krajcarz, M., Krokhaleva, M., Lebreton, L., Lemanik, A., Maul, L.C., Nagel, D., Noiret, P., Primault, J., Rekovets, L., Rhodes, S., Royer, A., Serdyuk, N.V, Soressi, M., Stewart, J., Strukova, T., Talamo, S., Wilczyński, J., Nadachowski, A. (2023a). Ancient DNA of narrow-headed vole reveal common features of the Late Pleistocene population dynamics in cold-adapted small mammals. *Proceedings of the Royal Society B 290*(1993), 20222238.

Brace, S., Ruddy, M., Miller, R., Schreve, D.C., Stewart, J.R., Barnes, I. (2016). The colonization history of British water vole (Arvicola amphibius (Linnaeus, 1758)): origins and development of the Celtic fringe. *Proceedings of the Royal Society B: Biological Sciences 283*(1829), 20160130.

Costamagno, S., Barshay-Szmidt, C., Kuntz, D., Laroulandie, V., Pétillon, J. M., Boudadi-Maligne, M., Langlais, M., Mallye, J.-B., Chevallier, A. (2016). Reexamining the timing of reindeer disappearance in southwestern France in the larger context of late glacial faunal turnover. *Quaternary International 414*, 34-61.

Mallye, J. B., Kuntz, D., Langlais, M., Boudadi-Maligne, M., Barshay-Szmidt, C., Costamagno, S., Pétillon, J.-M., Gourichon, L., Laroulandie, V. (2018). Trente ans après, que reste-t-il du modèle d'azilianisation proposé au Morin par F. Bordes et D. de Sonneville-Bordes? In *Table-ronde organisée en hommage à Guy Célérier" Les sociétés de la transition du Paléolithique final au début du Mésolithique dans l'espace nord aquitain".* Paléo No. spécial, 155-168.

Rofes, J., Cersoy, S., Zazzo, A., Royer, A., Nicod, P. Y., Laroulandie, V., Langlais, M., Pailler, Y., Leandri, C., Lebon, M., Tresset, A. (2020). Detecting stratigraphical issues using direct radiocarbon dating from small-mammal remains. *Journal of Quaternary Science 35*(4), 505-513.

Royer, A., Sécher, A., Langlais, M. (2018). A Brief Note on the Presence of the Common Hamster during the Late Glacial Period in Southwestern France. *Quaternary 1*(1), 8.

Woodman, P., McCarthy, M., Monaghan, N. (1997). The Irish quaternary fauna project. *Quaternary Science Reviews 16*(2), 129-159.

Table 1. I'm sure this would be worked out in production, but the lines and Zonobiome descriptions for Zones VII and VIII are confusing. Which zone does "Boreal coniferous forest" and (taiga) and "coniferous" belong to?

**The "Boreal coniferous forest" belongs to VIII. We have corrected the problem, which appears both because of the repetition of "coniferous" word (that we removed) and because of the table layout, which we have corrected.**

Line 226. I understand why the authors chose to focus only on temperature values – this is a big undertaking. But for future work, seeing how measures of precipitation perform would be valuable, since many small mammals are very sensitive to precipitation (likely through its effects on vegetation).

**Reconstruct precipitation is clearly one of the objectives for the future. The first step would be to update fossil data to include Eulipotyphla, which are highly sensitive to precipitation/aridity. However they have the disadvantage of having been less systematically analyzed than rodents in most of the fossil assemblages. Secondly, it would be probably important to evaluate these precipitation reconstructions in relation to temperatures, as considering this climate variable alone in the Bioclimatic Analysis gives less accurate reconstructions (R² around 0.77 in Royer et al., 2020).**

Section 2.3, and especially lines 215 – 229. There are two places where the bioclimatic model estimates values for extinct communities – it estimates the bioclimatic zone using linear discriminant functions, and it estimates different climate variables using transfer functions. For this second part, the authors write "The second part of the Bioclimatic Model is built from transfer functions by means of multiple linear regression analyses of climatic parameters and modern bioclimatic spectra. These models are ultimately used to infer climatic variables for additional observations (i.e., extinct communities)." The focus in these sentences is on inferring climate for past communities. But for the modern communities, how does model validation work? Is the model validated by leaving out sets of modern communities? Or have the transfer functions already been validated and even though there is new CRI data for some species, the authors rely on those same transfer functions? I understand the authors are building on other work here (e.g., Hernández Fernández & Peláez-Campomanes, 2005; Royer et al., 2020), but a few additional details would be helpful.

**The models used here were already validated in Royer et al. (2020), see modifications in the text below for details. Therefore, we rely here on the same transfer functions obtained in Royer et al. (2020). Although the species' CRIs published in that paper only cover the modern species living in the 49 modern communities used to generate the model, the addition of new species from modern communities (the 157 in this paper) and also of species found in the fossil sites does not modify the models. In any case, as already noted in the text, we used the same approach to generate the CRIs for all these new species.**

**We modified the text to clarify the point raised by the reviewer as follows (in bold the changes from the original text):**

**"The original models for the Bioclimatic Analysis (Hernández Fernández & Peláez-Campomanes, 2003, 2005) were based on 50 modern localities at global scale and successfully validated by using an additional set of different localities from the ones used to develop them. The current models constructed by Royer et al. (2020) were based on a**

**new set of 49 modern communities distributed throughout the Palearctic, in order to be representative of the different climate zones (seven localities for seven climate zones). In that paper, the models were also validated based on Leave-One-Out Cross Validation (LOOCV).**

[…]

The Bioclimatic **Analysis** is composed of two parts. The first one relies on linear discriminant functions deduced from the bioclimatic spectra of modern mammalian communities. These linear discriminant functions are subsequently used to classify additional observations (extinct communities in our case) in each climatic zone, with an associated posterior probability (Hernández Fernández & Peláez-Campomanes, 2003). We used posterior probability values to assess the robustness of the climate classifications obtained by the discriminant functions, and considered robust probabilities above 0.95. **A prediction error was estimated around 12% for estimating bioclimatic zone with linear discriminant functions on rodent communities (Royer et al., 2020).** The second part of the Bioclimatic **Analysis** is built from transfer functions by means of multiple linear regression analyses of climatic parameters and modern bioclimatic spectra. **Most predictive equations generated by multiple linear regressions for each climatic factor, produced highly significant determination coefficients (Hernández Fernández & Peláez-Campomanes, 2005), and rarefaction analysis revealed these new models to be reliable even when a substantial percentage of species from the original community was removed (Royer et al., 2020).** These models are ultimately used to infer climatic variables for additional observations (*i.e.* extinct communities). Although the Bioclimatic **Analysis** gives the possibility to estimate eleven climatic variables from fossil faunal assemblages (Hernández Fernández & Peláez-Campomanes, 2005; Royer et al., 2020), in this paper, we focus only on three of them: the Mean Annual surface Temperature (MAT), the Mean surface Temperature of the WArmest month (MTWA) and the Mean surface Temperature of the COldest month (MTCO), which are characterized by coefficients of determination of 0.94, 0.92 and 0.85, respectively (Royer et al., 2020). **"**

Lines 306-307. This sentence needs editing for clarity. I am not sure exactly what the authors are trying to say with the second part ("and the benefit of using the Bioclimatic Model to partially overcome these limitations"). I think they maybe mean "and using the Bioclimatic Model helps to partially overcome these limitations".

**That is what we meant. We have modified the sentence as suggested by the reviewer.**

Figure 2, and other places throughout the paper. The labels on Figure 2 indicate the "Biozone". But the term "biozone" only really is used in the figures. In other places (ie, the caption to figure 2, "3.2.1 Climate zone classification", many other places in the main text), the zones are called "climate zones". This led to confusion for me initially - I thought these were two separate things. I would try to use the same terminology throughout.

**We have homogenized the figures and the text, by replacing the word "biozone" by "climate zone".**

Figure 3, Fig 6, Fig 9, Fig 10, Fig 11, Fig 12, Fig 13. The site icons (triangles, squares, circles, etc) are very small, and difficult to see clearly without zooming in quite far. And the colors of the labels vs background makes it extra difficult for some figures. Thus, the authors should consider increasing the size of the site icons (such as in Figure 2) or perhaps outlining the icons in black – while the spatial location may appear less precise, at the scale of the study that level of precision is not necessary.

**We have modified the figures to enlarge the icons and add an outline to the icons.**

Figure 6 vs Figures 9b and Fig 12. In some cases, the legend in the box to the left of the maps on these figures does not reflect the map values. e.g., Fig 6 top row, it says "delta rodents – ERA5". I think this is accurate - the authors are mapping the difference (delta) between rodents and ERA5 [i.e., delta (rodents – ERA5)]. But in other cases (e.g., Fig 9b), the authors use the exact same notation ("delta rodents – Beyer2020" and "delta rodents – GCMs"), but in this case the caption indicates that what is mapped is "delta rodents - delta Beyer2020" and "delta rodents - delta GCMs". And same for Figure 12 - I think the label should be "delta rodents - delta Beyer2020". But perhaps I have gotten my interpretations wrong here. Regardless, I encourage the authors to critically examine the figure labels to ensure they all accurately reflect what is being mapped.

**We have modified the legends in the box to the left of Figure 9B and 12 to be in agreement with the map representations, as pointed out by the reviewer.**

Line 529, "This environment is far to be restricted to a tundra". This word choice is a bit unclear, so I'm not quite sure what the authors mean.

**We have deleted the misunderstood part of the sentence and modified the rest:**

**"This environment is most likely characterized by very fragmented forested cover and a scarcity of woodland refugia, which would be nonetheless associated to the occasional presence of some thermophilous species (Davis et al., 2022)."**

Line 547: "based on faunal species". Do you mean based on OTHER faunal species? Or perhaps "colder than OTHER estimations based on faunal species", if the Puzachenko and Markova estimates are from rodents as well.

**The Puzachenko and Markova estimation indeed comes from rodents. We have added the word 'other" in the sentence to clarify it.**

Figure 10 - some of the panels are offset from one another along the x axis. Are all panels spanning to 60°E longitude or do some of the panels/models end before then? If they are all the same, please re-size panels accordingly.

**There was some problems when extracting panel plots. We have fixed the problem, corrected and homogenized the plots.**

Line 580, "showing much LGM temperatures that are much colder". The first "much" can be deleted, I think.

**We have removed the first "much".**

Line 580, "zonal gradient'. I think this means "longitudinal", but it's not clear if it's supposed to mean that or the biozones/climate zones.

**We have changed the word "zonal" by "geographical", the gradient being both longitudinal and latitudinal.**

Line 668: "closed to" or "close to"? I think it's supposed to be the latter.

**We fixed the error, as well as the same one found line 635.**

Figure 13 - because the site labels are so small, it's difficult to tell apart circles from triangles. And when I zoom in, resolution issues also mean I can't easily differentiate.

**We have enlarged the symbols.**

Lines 693-694. What about the sea level rise? I think you should add "and the sea level rise [progressively eliminating some areas]".

**We have added this proposal as follows:**

**"The Late Glacial period experienced significant climatic shifts, marked by rapid climate oscillations, which led to a dynamic re-shaping of living spaces, with the progressive retreat of ice sheets freeing up new areas in the northern regions, while rising sea levels simultaneously submerged substantial portions of coastal regions."**

Line 707, "spatial pattern [different?] from Beyer2020".

**We forgot the word "different". We have fixed the mistake as follows: "These faunal changes reflect temperature changes that vary both in time and space, producing spatial pattern that differs from Beyer2020."**

Lines 718-720 if the rodent associations "generate colder temperature values in the western part of Europe and warmer in Eastern Europe" compared to GCMs, wouldn't there be a "[stronger] west-east temperature gradient"? I am unsure about the authors use of the word "weaker" in the original text.

**Error corrected: we should have used "stronger" instead of "weaker".**

Lines 721-722. What do you mean by "The physical interpretation …. is questionable"? Here, are you effectively saying it's not clear which scenario (rodents, GCMs) reflects real-world conditions during the LGM?

> Update: ah, I see. Yes, I think this is what you are saying, based on what is in the next paragraph. I think this would be more clear if you moved the sentence starting with "The physical interpretation" to be the start of the new paragraph, merged with lines 723 - 743.

**We have moved the sentence to the beginning of the next paragraph as proposed by the reviewer**

Line 751, "given niche species". Needs rewording. I think the authors mean the following: "underestimating the plasticity and adaptation of species niches with new abiotic and biotic constraints."

**We modified the sentence as follows: "The relationships between the species and their environment are in constant evolution, and the current observations reflect only a realized niche, potentially underestimating the plasticity and adaptive capacity of species in response to new abiotic and biotic constraints."**